# Risk Assessment of Progressive Multifocal Leukoencephalopathy in Multiple Sclerosis Patients during 1 Year of Ocrelizumab Treatment

**DOI:** 10.3390/v13091684

**Published:** 2021-08-25

**Authors:** Carla Prezioso, Alfonso Grimaldi, Doriana Landi, Carolina Gabri Nicoletti, Gabriele Brazzini, Francesca Piacentini, Sara Passerini, Dolores Limongi, Marco Ciotti, Anna Teresa Palamara, Girolama Alessandra Marfia, Valeria Pietropaolo

**Affiliations:** 1IRCSS San Raffaele Roma, Microbiology of Chronic Neuro-Degenerative Pathologies, 00163 Rome, Italy; 2Department of Public Health and Infectious Diseases, “Sapienza” University of Rome, 00185 Rome, Italy; gabriele.brazzini@uniroma1.it (G.B.); francescapiacentini.1854105@studenti.uniroma1.it (F.P.); passerini.1915659@studenti.uniroma1.it (S.P.); 3Multiple Sclerosis Clinical and Research Unit, Fondazione Policlinico di Tor Vergata, 00133 Rome, Italy; alfonso.grimaldi@uniroma2.it (A.G.); doriana.landi@gmail.com (D.L.); carolgabri@gmail.com (C.G.N.); marfia@uniroma2.it (G.A.M.); 4Department of Systems Medicine, Tor Vergata University, 00133 Rome, Italy; 5IRCCS San Raffaele Roma, Telematic University, 00163 Rome, Italy; dolores.limongi@uniroma5.it; 6Laboratory of Virology, Polyclinic Tor Vergata Foundation, 00133 Rome, Italy; marco.ciotti@ptvonline.it; 7Department of Infectious Diseases, Istituto Superiore di Sanità, 00161 Rome, Italy; annateresa.palamara@iss.it; 8Laboratory Affiliated to Institute Pasteur Italia-Cenci Bolognetti Foundation, Department of Public Health and Infectious Diseases, “Sapienza” University of Rome, 00185 Rome, Italy; 9Unit of Neurology, IRCCS Istituto Neurologico Mediterraneo NEUROMED, 86077 Pozzilli, Italy

**Keywords:** John Cunningham virus (JCPyV), progressive multifocal leukoencephalopathy (PML), multiple sclerosis (MS), ocrelizumab, JCPyV DNA-detection, NCCR re-arrangements, JCPyV serostatus

## Abstract

Background: Progressive multifocal leukoencephalopathy (PML) caused by the JC virus is the main limitation to the use of disease modifying therapies for treatment of multiple sclerosis (MS). Methods: To assess the PML risk in course of ocrelizumab, urine and blood samples were collected from 42 MS patients at baseline (T0), at 6 (T2) and 12 months (T4) from the beginning of therapy. After JCPyV-DNA extraction, a quantitative-PCR (Q-PCR) was performed. Moreover, assessment of JCV-serostatus was obtained and arrangements’ analysis of non-coding control region (NCCR) and of viral capsid protein 1 (VP1) was carried out. Results: Q-PCR revealed JCPyV-DNA in urine at all selected time points, while JCPyV-DNA was detected in plasma at T4. From T0 to T4, JC viral load in urine was detected, increased in two logarithms and, significantly higher, compared to viremia. NCCR from urine was archetypal. Plasmatic NCCR displayed deletion, duplication, and point mutations. VP1 showed the S269F substitution involving the receptor-binding region. Anti-JCV index and IgM titer were found to statistically decrease during ocrelizumab treatment. Conclusions: Ocrelizumab in JCPyV-DNA positive patients is safe and did not determine PML cases. Combined monitoring of ocrelizumab’s effects on JCPyV pathogenicity and on host immunity might offer a complete insight towards predicting PML risk.

## 1. Introduction

Progressive multifocal leukoencephalopathy (PML) is a fatal neurodegenerative disease caused by JC polyomavirus (JCPyV), a circular, double-stranded DNA virus, belonging to the *Polyomaviridae* family [1,2,3,4,5]. The JCPyV genome consists of early and late regions separated by a non-coding control region (NCCR), which is a key regulatory region harboring the origin of viral DNA replication ori, TATA-, TATA-like sequences, several transcription factors binding sites, promoter/enhancer elements, and the binding sites for the viral large T-antigen [6]. One remarkable feature of the JCPyV is the rearrangement of the promoter/enhancer elements of the NCCR [7,8,9,10], characterizing the virulent neurotropic pathogenic form (prototype), typically found in the cerebrospinal fluid (CSF), brain, and blood of PML patients. Conversely, the non-rearranged NCCR is associated to the non-pathogenic form (archetype), which is most frequently found in the urine of healthy individuals, and it is rarely found in the brain of PML patients [11]. In the human population, virus seroprevalence ranges from 50 to 60% and, despite the large proportion of individuals seropositive for JCPyV, PML represents a rare event. PML is always associated with severe weakening of the immune system, particularly when cell-mediated immune responses are involved. The first PML cases were described in patients affected by chronic lymphocytic leukemia and Hodgkin’s lymphoma [12]. Until the early 1980s, about 200 cases of PML were reported and all were associated with lymphoproliferative disorders [13]. Starting from the mid-1980s, with the HIV/AIDS epidemic, AIDS became the main PML risk factor with up to 5% of AIDS-related deaths associated with PML [14,15,16,17,18]. The introduction of effective antiretroviral therapy decreased the PML risk in HIV-infected patients to less than 1% [14,15,16,17,18] although PML remains a complication and cause of mortality in this population [19,20,21]. More recently, PML has reemerged as an opportunistic infection in other immunosuppressive conditions such as multiple sclerosis (MS) [22,23,24,25,26,27]. PML was first described in patients with MS in 2005 and it was attributed to the administration of natalizumab [28], a monoclonal antibody binding the very late antigen-4 (VLA-4) expressed on leukocytes [29]. The mechanism by which natalizumab increases the risk of PML is, to date, not yet understood, although it has been hypothesized that the low CNS immune surveillance and the increased presence of B cells and CD34+ progenitor cells could favor JCPyV replication. In 2004, natalizumab was released on the market and, when three cases of PML were identified, withdrawn [25,30]. In 2006, the drug was reintroduced after the establishment of a global risk-management protocol including, as specific risk factors, anti-JCPyV antibody status, the duration of natalizumab treatment, and prior immunosuppressant use [31]. At present, the risk-mitigation strategy, developed for natalizumab, is probably applicable only in relation with this drug. Among available and emerging disease modifying treatments (DMTs) for treatment of MS, long-term use of rituximab, a monoclonal antibody targeting CD20, has not been associated with a higher risk of PML in MS patients, despite many PML cases in the contest of hematological malignancies and other diseases with greater underlying risk of PML [32,33]. Treatment with Rituximab determines the destruction of peripheral B cells resulting in replacement by pre-B cells in bone marrow [34]. Long-term B-cell depletion reduces antibody production and modulates antigen presentation and activation of T cells and macrophages [35]. After over 10 years, the assessment of relationship between PML and rituximab is still uncertain. The current situation is complicated due to the approval of the first rituximab biosimilars and the introduction of other anti-CD20 monoclonal antibodies for which some signal of PML is already available [36]. Ocrelizumab, another recombinant humanized antibody, selectively targeting cells expressing CD20 on their surface, has recently been approved to treat relapsing remitting (RRMS) and primary progressive MS (PPMS). While long-term effects of rituximab have been associated with the development of PML at an incidence of 1/25,000 in non-MS conditions, the long-term use of ocrelizumab are essentially unknown [37]. To date, nine cases of PML have been reported during treatment with ocrelizumab, mostly (8/9) as carry-over cases from previous DMTs [38]. PML risk in the course of DMTs is lower (fingolimod: 0.069–0.082/1000 treated patients; dimethyl fumarate: 0.018/1000 treated patients; alemtuzumab: 1 case after second cycle of alemtuzumab, 3 suspected cases, 1 case with pre-exposure to natalizumab; cladribine: few cases, years after treated of hematological condition) than the risk associated with natalizumab (4.17/1000 treated patients), but it is not clear whether this is due to the different mechanism of action of these drugs or inappropriate screening methodology [39]. To date, PML risk-mitigating strategy for ocrelizumab is still lacking. An anti-JCV index is commonly used in clinical setting as marker of viral replication, which usually also takes place in the B lymphocytes reservoir, but, since ocrelizumab exerts its effect by depleting B lymphocytes from peripheral circulation, monitoring anti-JCV index may be unreliable to monitor individual risk of PML. Thus, given this background, in this study, the reliability of the anti-JCV index during one year of ocrelizumab treatment was evaluated among MS patients concomitantly with JCPyV replication and NCCR behavior. The identification and validation of a biomarker predictive of a severe adverse event such as PML is of paramount importance to guarantee the safety of DMTs such as ocrelizumab. Moreover, since ocrelizumab may be used in patients switching from other drugs to improve PML risk assessment and tailor better prevention strategies, it represents an obligatory requirement of the MS treatment algorithm. Combined monitoring of the ocrelizumab’s effects on JCPyV pathogenicity and on host immunity might offer and complete insight towards predicting risk of PML.

## 2. Materials and Methods

### 2.1. Study Participants and Samples Collection

Following the approval of the study by the Independent Ethic Committee of Policlinico Tor Vergata (protocol number 8473/2019), a signed informed consent was obtained from 42 patients diagnosed with a relapsing–remitting multiple sclerosis (RRMS) enrolled in the study between February 2019 and May 2020. The study group was composed of 19 patients naive to MS treatment, 4 patients already receiving natalizumab, and 19 patients switching from other treatments, specifically fingolimod, dimethyl fumarate, and Aubagio. Throughout the study period, all patients received ocrelizumab-based treatment. Therapeutic protocol consisted in the intravenous administration of 600 mg of ocrelizumab 2 weeks apart, followed by a single 600-mg intravenous infusion every 6 months. From the enrolled patients (18 males and 24 females; mean age ± standard dev. 40.83 ± 9.33), 42 plasma and 42 urine samples were obtained before ocrelizumab treatment (baseline T0: 0 infusions) and, after beginning of therapy, at the following time points: T1, 3 months, T2, 6 months, T3, 9 months, and T4, 12 months, for a total of 420 specimens. Demographic and clinical characteristics are presented in Table 1. Neurological examination including Expanded Disability Status Scale (EDSS) scoring was performed at T0 and then every 12 months (T4) [40].

### 2.2. Virological Investigations

JCPyV DNA was extracted using DNeasy Blood & Tissue Kit (QIAGEN, Milan, Italy). Extraction products were analyzed by a quantitative PCR (Q-PCR) system able to detect a 54-bp amplicon in JCPyV T antigen region, using a 7300 real-time PCR system (Applied Biosystems, Waltham, MA, USA) [41,42]. Each sample was analyzed in triplicate, and JCPyV DNA loads (given as the mean of at least three positive reactions) were expressed as genome equivalents (gEq)/milliliter. Negative and positive controls were included in each Q-PCR session. The standard curve was obtained from serial dilutions (range: 10^5^–10^2^ gEq/mL) of a plasmid containing the entire JCPyV genome. The lower detection limit of the Q-PCR system was 10 DNA copies of the target gene per amplification reaction, corresponding to 10 genome equivalents per reaction (10 gEq/reaction). JCPyV DNA positive samples were further analyzed using nested-PCR for NCCR and VP1 regions’ amplification [43,44,45]. PCR products were analyzed on 2% agarose gels. The amplified products were purified using a MinElute PCR Purification Kit (QIAGEN, Milan, Italy) and sequenced in a dedicated facility (Bio-Fab research, Rome, Italy). Obtained sequences were compared to reference strain (GenBank: AB081613). Sequence alignment was performed using ClustalW2 [46] on the European Molecular Biology Laboratory–European Bioinformatics Institute (EMBL–EBI) website using default parameters. JCPyV genotypes/subtypes were classified based on single nucleotide polymorphisms (SNPs) found within the amplified VP1 region [47]. VP1 phylogenetic analysis was carried out using representative strains obtained from urine and plasma compared with JCPyV reference strain AB081613. Alignment was performed with ClustalW2, and a phylogenetic tree was created using Molecular Evolutionary Genetics Analysis (Mega), using a neighbor-joining algorithm [48]. A bootstrap test with 1000 replicates was performed to evaluate the confidence of the branching pattern of the trees.

### 2.3. Serological Investigations

Assessment of immunoglobulin (IgM and IgG) titer was determined on serum of enrolled patients at each visit from T0 e during throughout the follow up (T1–T4). Furthermore, risk stratification was performed by using an anti-JCV index level, using a commercially available enzyme-linked immunosorbent assay (ELISA), considering 1.5 as the cut-off value. Consequently, three risk categories, low (≤0.9), intermediate (0.9 < JCV index > 1.5), and high (>1.5) were distinguished. The change in absolute CD19 and CD20 cells count was investigated to determine the effectiveness of treatment with B-cell suppression.

### 2.4. Statistical Analysis

The results were analyzed with the use of the statistical method. The continuous variables were expressed both as mean ± SD and as median and range. All studied features were analyzed with non-parametric test, such as the Friedman chi-square test, Kruskal–Wallis test, and Mann-Whitney U for unmatched data. This due to the result of Shapiro–Wilk test which showed a non-normal distribution of data. The statistically significant *p*-value level was set at <0.05.

## 3. Results

### 3.1. JCPyV DNA in Urine and Plasma at Different Follow-Up Times

Each sample (42 urine and 42 plasma for a total of 84 specimens for each time point) was tested for the presence of JCPyV DNA. Overall, at T0 (before starting ocrelizumab treatment), among the enrolled population (19 naïve patients, 4 patients switching from natalizumab, and 19 patients previously treated with other drugs), JCPyV DNA was detected in 34/42 (81%) urine and in 0/42 plasma samples. Specifically, viral DNA was detected in 11 out of 19 naive patients, in 4 out of 4 natalizumab patients, and in 19 out of 19 patients switching from other treatments for MS. Analysis of qPCR showed a viral load ranged from 4 × 10^4^ copies/mL to 1.1 × 10^6^ copies/mL, mean value of 7.6 × 10^5^ copies/mL, and median cycle threshold value (Ct) of detection of 25.18 (interquartile range (IQR) 22.02–28.18).

At T2 (six months of ocrelizumab administration), JCPyV DNA was detected in 34/42 (81%) urine and in 0/42 plasma samples. Viral DNA was detected in 11 out of 19 naive patients, in 4 out of 4 natalizumab patients, and in 19 out of 19 patients switching from other treatments. Results of qPCR revealed an increase in viral load that ranged from 1.2 × 10^5^ to 5.8 × 10^7^ with a mean value of 5.3 × 10^6^ copies/mL and median Ct value of detection of 24.36 (IQR) 20.08–26.07.

Finally, at T4 (12 months of ocrelizumab infusion), the number of patients presenting JCPyV DNA in urine was 35/42 (83%). Viral DNA was revealed among 12 out of 19 naive patients, in 4 out of 4 natalizumab patients, and in 19 out of 19 patients switching from other treatments. Analysis of qPCR showed a further increase in viral load that ranged from 1 × 10^6^ to 9 × 10^9^ with a mean value of 3.8 × 10^7^ copies/mL and median Ct of detection of 24.14 (IQR 21.010–27.32). Plasma samples were confirmed negative on viral detection except for three patients, in which viruria was revealed (3/42) with a viral load ranged from 1.5 × 10^2^ to 2.0 × 10^3^ with a mean value of 6.5 × 10^2^ copies/mL and median Ct value of detection of 28.32 (IQR) 23.15–29.40). These patients were simultaneously positive in plasma and in urine and were previously treated with natalizumab (1/4) and other drugs (2/19) (Table 2).

### 3.2. NCCR and VP1 Analysis

From T0 to T4 JCPyV NCCR analysis showed, as expected, a structural organization identical to archetype JCPyV in all analyzed positive urine samples (Figure 1A). The NCCR analysis performed at T4 on the 3 positive plasma samples displayed mutations throughout the NCCR sequence. Specifically, the first plasmatic NCCR analyzed, was characterized by the transversion 37T→G in Spi-B binding site (box B), by a duplication of NF-1 cellular transcription factor binding site (box F) and by a box D composed only of 10 out of 65 bases (nucleotides from 117 to 126). The resulted NCCR was A– B*–C–(D)–E–F* (Figure 1B). VP1 coding region was also sequenced and showed a wild-type structure without any mutations. The second JCPyV NCCR sequenced, revealed the following block organization: A–B*–C–C–E–F (Figure 1C). The transversion 37T→G in Spi-B binding site (box B) was observed together with duplication of box C and deletion of nucleotides from 117 to 180, corresponding to box D (Figure 1C). VP1 region sequencing revealed a single mutation causing the amino acid change S267L involving the receptor-binding region of VP1. In detail, S267L was related to 1A genotype. Finally, the third JCPyV NCCR studied showed the transition 217G→A in box F (Figure 1D). Sequencing of VP1 coding region presented a wild-type structure without mutations.

### 3.3. Phylogenetic Analysis

Phylogenetic analysis, carried out on VP1 sequences isolated from urine and plasma samples, revealed that all isolates were 99.9% similar to JCPyV reference strain AB081613 (CY strain). Figure 1: NCCR structure analysis during one year of ocrelizumab treatment.

### 3.4. Assessment of Immunoglobulins and Anti-JCPyV Index in MS Patients

Immunoglobulin levels of IgM and IgG was investigated both prior to start the ocrelizumab treatment (T0) and during ocrelizumab administration (T2 and T4). The IgM titer decreased during ocrelizumab treatment with a statistically significant decrease in mean values from 127.95 (± 53.50) at T0 to 75.04 (± 41.93) at T4 (*p* < 0.05) (Table 3). IgG levels of pre-ocrelizumab (T0) and during ocrelizumab administration had a stationary trend over time (*p* > 0.05) (Table 3). Four months after starting ocrelizumab, as expected, there was a statistically significant decrease in the CD19 absolute cell count (*p* < 0.05). Specifically, 34/42 patients were totally suppressed (81%), whereas 8/42 (19%) had a CD19+ cell count ranging between 0.1% and 0.9%. These data confirm that patients appropriately responded to treatment with B-cell suppression. No patient with a significant increase of CD19+ cells was observed.

For comparison, during the study’s observation period, the anti-JCV index value pre- and post-initiation of ocrelizumab treatments was also measured.

At T0, patients who had an anti-JCV index > 1.5 were 26/42. Specifically, 4/19 were naive patients, 3/4 were patients switching from natalizumab, and 19/19 were patients previously treated with other drugs.

At T2, the number of patients presenting an anti-JCV index > 1.5 was confirmed the same observed at T0: 26/42 (4 naive, 3 switching from natalizumab and 19 were previously treated with other drugs).

Finally, at T4, patients who had an anti JCV index > 1.5 were 27/42 (4 naive, 4 switching from natalizumab and 19 patients previously treated with other drugs (Table 4).

### 3.5. Combined Monitoring of the Serostatus and Its Relationship to Viral DNA

At point of follow-up T0, patients who had an anti-JCV index >1.5 were 26/42. Among these, patients presented concomitantly viruria. Analysis of qPCR results showed an amount of JCPyV DNA with a mean value of 1.0 × 10^6^ copies/mL. At T2, patients presenting an anti-JCV index >1.5 remain 26 (26/42) and JC viruria was detected in each of these patients, whereas, at T4, the number of those showing an anti-JCV index >1.5 was 27 (27/42). These patients presented simultaneously JCPyV DNA in urine (Table 5). Analysis of qPCR results showed an amount of JCPyV DNA with a mean value of 9.0 × 10^6^ copies/mL at T2 and of 8.8 × 10^7^ copies/mL, at T4 (Table 5). It is worth noting that, at T4, 1 patient switching from natalizumab, and 2 patients switching from other drugs, with an anti-JCV index >1.5 also presented, in addition to viruria, viremia with a JCPyV DNA mean value of 1.5 × 10^3^ copies/mL.

There was one patient presenting a medium risk of PML (0.9 < JCV index > 1.5) at T0 and at T2 (1/42), who was previously treated with natalizumab (Table 5). At T4, a different patient had an anti-JCV index between 0.9 and 1.5 and was a naive patient. Both patients excreted JCPyV DNA from urine with a JCPyV DNA amount of 7.5 × 10^4^ copies/mL and of 1.5 × 10^5^ copies/mL at T0 and T2, respectively. A naive patient, at T4, presented a viral DNA amount of 2.0 × 10^3^ copies/mL (Table 5).

Finally, patients with an anti-JCV index ≤ 0.9 were 15 at T0 (15/42), was confirmed to be 15 at T2 (15/42) and decreased to 14 at T4 (13/42). At T0, MS patients presenting JCPyV DNA were 15, at T2 were 15, whereas, at T4, patients presenting viral DNA decreased to 14 in agreement with an anti-JCV index. These patients were naive MS patients. Analysis of qPCR results showed an amount of JCPyV DNA with a mean value of 3.0 × 10^3^ copies/mL at T0. At T2, qPCR results displayed a JCPyV DNA mean value of 9.8 × 10^3^ copies/mL and of 1.0 × 10^4^ copies/mL at T4 (Table 5).

## 4. Discussion

PML is a rare CNS disease caused by a lytic infection of oligodendrocytes due to the reactivation of JC virus. It can be associated with MS-DMTs, including natalizumab and less frequently with fingolimod and dimethyl-fumarate. B-cell depleting therapy with rituximab has also been associated with PML [49]. Possible PML risk also exists for ocrelizumab, given the structural similarity to the chimeric anti-CD20 monoclonal antibody rituximab [49]. To date, nine cases of PML have been reported during ocrelizumab treatment, 8 cases out of 9 as carry-over from previous DMTs [38]. Anti-JCV seropositivity and anti-JCV antibody index, determined by the two-step second generation STRATIFY JCV^TM^ ELISA, are validated tools to stratify PML risk during natalizumab therapy [50,51]. It has been demonstrated that, in course of natalizumab, there is a rise in JCPyV seropositivity and in anti-JCV antibody index associated with an increased risk for PML [52]. Although an anti-JCV antibody index is employed to assess PML risk also in course of other treatments, its usefulness is unproven. In fingolimod-treated patients, a decrease in anti-JCV antibody index has been reported, and it has been attributed to a decrease in numbers of circulating lymphocytes [53]. A decrease in anti-JCV antibody index and immunoglobulins has also been reported in MS patients treated with rituximab [37], in contrast with a retrospective study reporting an increase in anti-JCV antibody titers during ocrelizumab-treatment in MS patients [54].

Due to conflicting results, the failure of validation of the anti-JCV antibody index, as a tool to stratify PML risk during treatment with ocrelizumab, and the requirement to identify and validate biomarker predictive of a severe adverse event such as PML, in this study previous data were deepened regarding the assessment of anti-JCV antibody index in MS patients, during one year of ocrelizumab treatment, also considering the ocrelizumab’s effects on JCPyV pathogenicity in terms of viral replication and NCCR behavior.

Regarding the trend of JCPyV viruria and viremia at different follow-up times, our results showed that JC viral DNA was detected in urine for the whole study (T0–T4) and in plasma after one year of ocrelizumab administration (T4). Specifically, viral DNA was detected at T0 in 58% of naive patients (11/19), in 100% of patients previously treated with natalizumab (4/4) and in 100% of patients switching from other treatments for MS (19/19; fingolimod, dimethyl fumarate, and Aubagio).

At T0, JC viruria was observed in naive patients, confirming that JC shedding occurs in approximately 13–20% of healthy individuals, independently from the administration of different immunomodulatory medications or the presence of several immunocompromising factors related to their underlying disease.

Interestingly, at T2 the number of patients presenting JC viruria was the same observed at T0 endorsing that ocrelizumab treatment did not affect “viral-conversion”. Finally, at T4 (12 months after the beginning of therapy), the number of naive patients increased from 11 to 12 (12 out of 19 naive patients, in 4 out of 4 natalizumab patients, and in 19 out of 19 patients switching from other treatments). Although the number of patients with JC viruria remained almost constant during the study, JC viral load in urine increased of two logarithms from T0 to T4, and, significantly higher, compared to viremia revealed at T4. This result could be explained assuming that, ocrelizumab, in addition to its effect on B-cells, might influence T-cell immunity, predisposing to viral and opportunistic infections and reactivation as previously described also for rituximab [55]. Moreover, in addition to the role in antibody synthesis, B-cells may also act as antigen-presenting cells, contributing to the dysregulation of cofactors involved in an effective immune response. Ocrelizumab might be able to induce immunosuppression through other mechanisms such as neutropenia, especially when administered for long periods [56]. In our study, it appears reasonable that ocrelizumab might have contributed to JCPyV infection/reactivation, although no definite conclusions can be drawn. It is possible assume that the decrease in immunosurveillance may have determined, firstly, a JCPyV replication in renal cells with a viral shedding and, as expected from JCPyV biology, a subsequent release of JCPyV infected cells into bloodstream [56]. Because JC viremia could be related with the onset of neuro-pathogenic JCPyV, monitoring the trend of JC viral load could be a useful tool to identify mutations and/or NCCR re-organization, correlated with neuro-virulent variants [27]. At T4, the mean value of JCPyV viruria was of 3.8 × 10^7^ copies/mL and JCPyV DNA was concomitantly detected in plasma samples of three patients (7%) previously treated with natalizumab and other drugs (fingolimod and Aubagio).

The analysis of JCPyV NCCR showed a structural organization identical to archetype JCPyV in all MS patients’ urine samples throughout the study, while the NCCR analysis, performed on the 3 positive plasma samples, displayed mutations on NCCR sequence. Specifically, the first plasmatic NCCR analyzed was characterized by a duplication of NF-1 cellular transcription factor binding site (box F) and by a box D composed only of 10 out of 65 bases (nucleotides from 117 to 126). It is well known that NF-1 increased the expression of JCPyV’s early and late genes, promoting onset of JCPyV variants with determinants of neurotropism and increased neurovirulence [57]. The second JCPyV NCCR sequenced revealed the following block organization: A–B*–C–C–E–F. Moreover, the transversion 37T→G in Spi-B binding site (box B) was observed together with duplication of box C and deletion of nucleotides from 117 to 180, corresponding to box D. Duplication of box C corresponds to a duplication of the CRE element binding site, in which resides the potential neurovirulence conferred by the CRE element, a specific enhancer of JCPyV replication [58], whereas the deletion of D box could represent one of the early but crucial steps in the complex series of NCCR rearrangements leading to PML. In our study we confirmed that the NCCR box duplications occur often near to the origin of genome replication (left box A to C), whereas deletions were more frequent near to late genes (right box D to F) [57,59]. Duplications in the left end of NCCR could confer a gain of function, and increase viral replication and gene transcription, whereas deletions, near to late genes, could determine a loss of function removing and suppressing control sequence [57]. Finally, the third JCPyV NCCR studied showed the transition 217G→A in Spi-B binding site of box F. It has been described that this transition could precede NCCR re-organization, and it is able to transform archetype Spi-B binding site in a JCPyV PML-variant [60].

VP1 coding region was also sequenced and, although *VP1* is considered a gene highly polymorphic, only one plasma sample showed a mutation causing the amino acid change S267L involving the receptor-binding region of VP1. This mutation could change the virus’s binding properties to its receptor or drive to alternative receptor usage [61]. Moreover, it was related to 1A genotype that represents the predominant genotype among white people of European descent, as predictable for our cohort of patients [62]. Finally, our data regarding VP1 phylogenetic analysis showed that all MS sequences isolated, were similar (99,9%) to the JCPyV reference strain AB081613 and fell into the same cluster. Although nucleotide mutations in structural *VP1* genes of other human polyomaviruses (HPyVs) have been described [62,63,64], in our study we detected only one change in the nucleotide order of the VP1 sequence. This result suggests a stability and a similarity across different isolates of the JCPyV VP1.

Since a higher JC viruria could correlate with viremia, as observed at T4, monitoring the trend of JC viral load could be a useful alert to identify mutations and/or NCCR re-organization. Although JCPyV NCCR arrangements and VP1 mutations are independent events impacting different elements of JCPyV cellular tropism, our results point to consider NCCR and VP1 modifications as biomarker to monitor the onset of neurovirulent variants since VP1 mutations have not been observed in archetype JCPyV NCCR, but only arise in JCPyV with the rearranged NCCR forms.

Regarding the ocrelizumab’s effects on host immunity, in this study the assessment of immunoglobulins and anti-JCPyV index was also evaluated. We found that 1-year use of ocrelizumab leads to decreases in immunoglobulin levels and particularly, a significant decrease was observed for IgM. This decrease has also been found during rituximab administration in the course of other conditions such as lymphoma and autoimmune diseases [65]. The decrease in immunoglobulins could potentially increase susceptibility to infections including infection/reactivation of JCPyV. For comparison, during the study’s observation period, the anti-JCV index value pre- and post-initiation of ocrelizumab treatments was also measured.

A decrease in mean value was observed over time, resulting statistically significant. It is possible to speculate that change of JCPyV serostatus is, at least partially, due to the natural fluctuation of antibody levels among individuals near to the cut-off point of the assay (1.5 cut-off value), or that a declining JCPyV index could represent the effects of ocrelizumab on circulating antibodies and likely does not correlate with a patient’s risk for PML.

Since previously different anti-JCV antibody index categories (low (≤0.9), intermediate (0.9 < JCV index > 1.5), and high (>1.5)) have been defined for PML risk stratification, due to this important clinical application, we have also applied these categories and investigated in which groups patients fell and if they switched between these different index categories during the follow-up study. At T0, patients who had a high anti-JCV index (>1.5) were 26/42 as well as at T2, whereas, at T4, we observed a patient previously treated with natalizumab that, from an anti-JCV index between 1.5 and 0.9, dropped to this group causing an increment from 26 to 27. A naive patient at T4, moves from a group characterized by an anti-JCV index ≤0.9 to a group in which an anti-JCV index was between 1.5 and 0.9. Additionally, in this single case it is possible to assume that patients can convert from high risk to intermediate or low and vice versa, as antibody concentrations normally fluctuate over time. Finally, patients presenting an anti-JCV index ≤ 0.9 were naive throughout the study and reflected the findings from sero-epidemiological studies in which it is estimated that more than 50% of the adult population worldwide have been exposed to JCPyV infection [66,67]. However, as reported from MS patients in treatment with natalizumab, in patients who have a low risk of PML, repeated measurements of anti-JCV index are necessary to exclude the possibility of seroconversion during treatments, since cases of PML were also reported in patients who had a low risk for PML, a few weeks before symptoms of JCPyV infection [68,69].

To further analyze the viral and antibody context of patients on ocrelizumab therapy for one year (T0–T4), a contextual analysis of JC viral load versus anti-JCPyV antibodies, was carried out. The results of the combined monitoring confirmed that the highest mean of JCPyV-DNA was revealed in patients presenting the higher value of an anti-JCV index (>1.5). Naive patients presenting an anti-JCV index ≤ 0.9 expelled JCPyV virions, but with a lower viral load. The patients with an intermediate risk of PML was at T0 and at T2 previously treated with natalizumab and at T4 was a naive patient. The higher amount of JCPyV DNA was observed in natalizumab patient in respect to the naive patient. The presence of such a numerous group of patients with an anti-JCV index >1.5 points out a clinical significance for treatment planning. The anti-JCV index alone does not significantly impact the risk of PML development, and we should keep in mind that most of these patients previously received natalizumab or other treatments such as fingolimod and dimethyl fumarate. In conclusion, the novelty of this study concerns the evaluation of the effects of the ocrelizumab therapy on viral replication and on NCCR behavior, concomitantly with the anti-JCV index. Although JCV-antibody index is commonly used in clinical setting as marker of JCPyV replication, we do believe that the metrics used to predict risk of PML could be implemented if viral load, viral genetic strain, and host systemic immunity are tacked together into account by neurologists and physicians from other fields. Through regular re-evaluation of these different parameters and an appropriate surveillance, an optimal PML mitigation strategy could be developed. A further unique aspect of this study is the fact that it was also conducted on naive MS patients, which allowed us to exclude the effect of MS previously therapies on the sero and viral status. Finally, despite the small sample size, this study could be considered innovative since it helps to provide an information on the incidence and prevalence of PML with respect to the underlying diseases (MS) and treatment (ocrelizumab). Our results showed that ocrelizumab therapy in JCPyV-DNA-positive patients is safe and did not determine PML cases. Often, studies on PML epidemiology and incidence trends are largely unknown and partly conflicting. The relatively short observational period limits the interpretation of the data but, follow-up study as this, extended for longer periods, could improve the risk–benefit ratio of the DMTs administration tailoring better prevention strategies, an obligatory requirement of MS treatment algorithm.

## Figures and Tables

**Figure 1 viruses-13-01684-f001:**
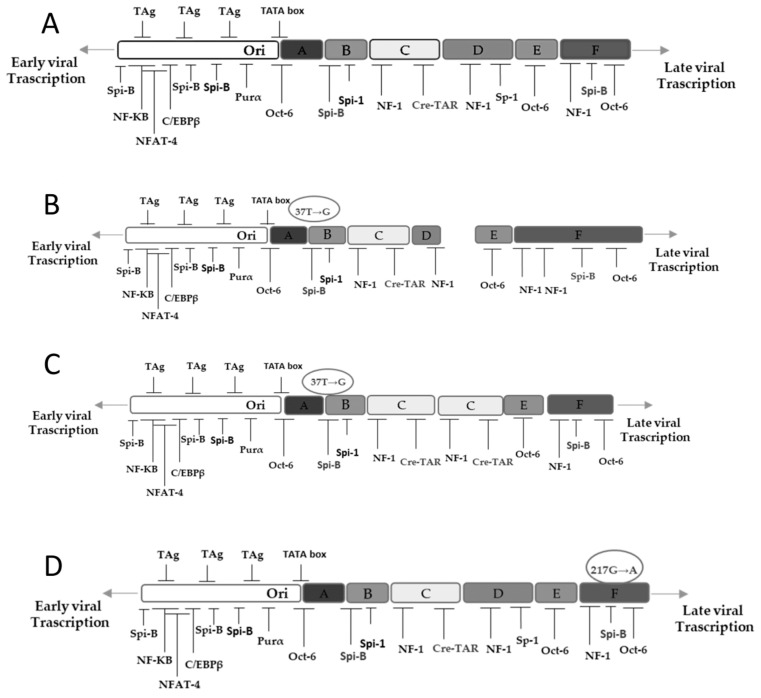
(**A**) Archetype JCPyV NCCR structure, observed in all MS patients’ urine samples was shown. (**B**) At T4 a plasmatic NCCR was characterized by 37T→G transversion in box B, inside Spi-B binding site, by a duplication of NF-1 cellular transcription factor binding site in box F and by a box D composed only of nucleotides from 117 to 126. (**C**) NCCR sequence isolated from plasma sample showed the 37T→G transversion in box B (Spi-B binding site), the duplication of box C, and the deletion of nucleotides from 117 to 180, corresponding to box D. (**D**) NCCR obtained from plasma displayed the transition 217G→A in box F.

**Table 1 viruses-13-01684-t001:** Demographic and clinical characteristics of MS population.

Features	Population		
Patients, *n*	42		
Sex, *n* (M/F)	**M**	**F**	
	18 (42.8%)	24 (57.1%)	
Mean age, years (SD)	40.34 (± 9.01)		
Median age, years (Range)	40.5 (56.5–22.0)		
Diagnosis, *n*	**RRMS**		
	42		
Pre-treatments, *n*	**Naive**	**Natalizumab**	**Others ***
	19	4	19

RRMS: relapsing–remitting multiple sclerosis; SD: standard deviation; * Others: fingolimod, dimethyl fumarate, and Aubagio.

**Table 2 viruses-13-01684-t002:** Viruria and viremia in ocrelizumab treated MS patients.

Features	T0	T2	T4
Positive viruria, *n*	34/42 (81%)	34/42 (81%)	35/42 (83%)
Range, *copies/mL*	1.1 × 10^6^ 4 × 10^4^	5.8 × 10^7^ 1.2 × 10^5^	9 × 10^9^ 1 × 10^6^
Mean viruria, *copies/mL*	7.6 × 10^5^	5.3 × 10^6^	3.8 × 10^7^
Positive viremia, *n*	0/42 (0%)	0/42 (0%)	3/42 (7%)
Range, *copies/mL*	0	0	2.0 × 10^3^ 1.5 × 10^2^
Mean viremia, *copies/mL*	0	0	6.5 × 10^2^

N: number of patients; T0: before starting ocrelizumab therapy; T2: 6 months after beginning of therapy; T4: 12 months after beginning of therapy.

**Table 3 viruses-13-01684-t003:** Immunoglobulins in MS patients.

	Mean T0 (SD)	Mean T2 (SD)	Mean T4 (SD)	F
**IgG**	960.08 (±202.22)	964.22 (±180.05)	978.14 (±191.09)	*p* > 0.05
**IgM**	127.95 (±53.50)	98.19 (±55.57)	75.04 (±41.93)	*p* < 0.05

F: Friedman Chi-square test; SD: standard deviation; T0: before starting ocrelizumab therapy; T2: 6 months after beginning of therapy; T4: 12 months after beginning of therapy.

**Table 4 viruses-13-01684-t004:** JCV index in MS patients.

Features	T0	T2	T4	F
JCV index ≥ 1.5, *n*	26	26	27	
0.9 < JCV index > 1.5, *n*	1	1	1	
JCV index ≤ 0.9, *n*	15	15	14	
Mean JCV index (SD)	2.24 (±1.53)	1.78 (±1.45)	1.56 (±1.38)	*p* < 0.05

T0: before starting ocrelizumab therapy; T2: 6 months after beginning of therapy; T4: 12 months after beginning of therapy; F: Friedman Chi-square test; SD: standard deviation.

**Table 5 viruses-13-01684-t005:** Analysis of viruria and viremia in patients with an anti-JCV index >1.5, 0.9 < JCV < 1.5 and <0.9.

	T0	T2	T4
JCV index > 1.5, *n*	26/42	26/42	27/42
Positive viruria, *n*	26/34	26/34	27/35
Mean viruria, *copies/mL*	1.0 × 10^6^	9.0 × 10^6^	8.8 × 10^7^
Positive viremia, *n*	0	0	3
Mean viruria, *copies/mL*			1.5 × 10^3^
0.9 < JCV < 1.5	1/42	1/42	1/42
Positive viruria, *n*	1/34	1/34	1/35
Viruria load, *copies/mL*	7.5 × 10^4^	1.5 × 10^5^	2.0 × 10^3^
Positive viremia, *n*	0	0	0
JCV ≤ 0.9	15/42	15/42	14/42
Positive viruria, *n*	15/34	15/34	14/35
Mean viruria, *copies/mL*	3.0 × 10^3^	9.8 × 10^3^	1.0 × 10^4^
Positive viremia, *n*	0	0	0

N: number of patients; T0: before starting ocrelizumab therapy; T2: 6 months after beginning of therapy; T4: 12 months after beginning of therapy.

## Data Availability

Data is contained within the article.

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
