# Peer review of "Risk Assessment of Progressive Multifocal Leukoencephalopathy in Multiple Sclerosis Patients during 1 Year of Ocrelizumab Treatment"

_viruses, 2021, doi:10.3390/v13091684_

Round 1
Reviewer 1 Report
The authors describe their work on the risk assessment of progressive multifocal leukoencephalopathy in mutiple sclerosis patients during 1 year of ocrelizumab treatment. This is an interesting study. Appropriate methodology has been employed and the conclusions appear to be justified based on the data at hand. However, I have a few recommendations for consideration.
- Introduction. This section could be shortened and more concise. Also in this section, the authors should provide a clear hypothesis to be tested in the study.
- Introduction. Can the authors provide some information on the incidence/prevalence of PML.
- Discussion. Does MS increase the risk for PML and are certain population groups at higher risk for PML.
- Discussion. Although the authors touch on the clinical applicability of their findings, in my opinion, the authors should emphasize and elaborate on this aspect.
- Discussion. Can the authors expand on the novelty aspect of their work.
Author Response
I submit a file with a point-by-point response to the reviewer’s comments
The authors describe their work on the risk assessment of progressive multifocal leukoencephalopathy in multiple sclerosis patients during 1 year of ocrelizumab treatment. This is an interesting study. Appropriate methodology has been employed and the conclusions appear to be justified based on the data at hand. However, I have a few recommendations for consideration.
Introduction. This section could be shortened and more concise. Also in this section, the authors should provide a clear hypothesis to be tested in the study.
Thank you for your suggestion. We shortened the introduction making this section more concise. Moreover, we tried to make the hypothesis of the study clearer.
Introduction. Can the authors provide some information on the incidence/prevalence of PML.
Thank you, we add the incidence/prevalence of PML during natalizumab treatment and also during other drugs.
Discussion. Does MS increase the risk for PML and are certain population groups at higher risk for PML.
Thank you. The prerequisite for PML is the profound suppression of cell-mediated immunity, whether associated with diseases, such as HIV or lymphoproliferative malignancies, or treatment with immunosuppressive or immunomodulatory therapies (multiple sclerosis or rheumatoid arthritis) or both (systemic lupus erythematosus). During the 1980’s and 1990’s with the emergence of HIV in humans, PML was the most important opportunistic infection of the CNS in these patients and most PML cases occurred in this group. In recent times, PML has been increasingly diagnosed in patients treated with biological therapies such as monoclonal Antibodies (mAbs) which deplete lymphocytes or impede leukocytes trafficking into the Central Nervous System. Interest in PML increased in 2005 when its association with the multiple sclerosis (MS) drug natalizumab was discovered and MS patients have become an important population at possible risk of PML. Natalizumab provides a good example how the risk-benefit profile must be determined at the individual level. PML risk also affects certain other MS therapies such as disease modifying therapies (DMTs). Population-based studies on PML epidemiology in course of MS are scarce and long-term overall incidence trends are largely unknown. Most papers report incidence in specific patient populations because PML was rare among patients not infected with HIV up to the mid-2000s or that are under mAb therapies. The discrepancy in incidence trends may be explained by differences in national circumstances and research methods. Nevertheless, they indicate that more studies from different regions and with in-depth analysis of PML causes are needed. Moreover, national PML surveillance and recording protocols would be helpful. The nature of immunosuppression underlying PML determines the prognosis, treatment, and the risk of PML immune reconstruction inflammatory syndrome (PML-IRIS). Detection of PML at an early stage, when the disease is asymptomatic and restricted, could be associated with a better outcome and higher survival rate. Cases of PML should be evaluated according to predisposing factors, as these subgroups differ by incidence rate, clinical course, and prognosis.
Discussion. Although the authors touch on the clinical applicability of their findings, in my opinion, the authors should emphasize and elaborate on this aspect.
Discussion. Can the authors expand on the novelty aspect of their work.
Thank you for the suggestions, we rewritten the conclusions of the discussion in a more incisive way.
Reviewer 2 Report
The present manuscript from Prezioso et al, explores the risk of progressive multifocal leukoencephalopathy in Multiple Sclerosis (MS) patients under ocrelizumab treatment. The safety of ocrelizumab in MS patients, including the viral replication and non-coding control region (NCCR) behavior concomitantly with the anti-JCV index in urine and blood samples, from 42 MS patients at T0, 6, and 12 months of therapy.
I just suggest checking the text again to detect and correct possible typographical or grammatical errors and Include in the legend of tables 2, 3, 4, and 5 the meaning of T0, T2, and T4.
Author Response
I submit a point-by-point response to the reviewer’s comments
The present manuscript from Prezioso et al, explores the risk of progressive multifocal leukoencephalopathy in Multiple Sclerosis (MS) patients under ocrelizumab treatment. The safety of ocrelizumab in MS patients, including the viral replication and non-coding control region (NCCR) behavior concomitantly with the anti-JCV index in urine and blood samples, from 42 MS patients at T0, 6, and 12 months of therapy.
I just suggest checking the text again to detect and correct possible typographical or grammatical errors and Include in the legend of tables 2, 3, 4, and 5 the meaning of T0, T2, and T4.
Thank you for your suggestion. We corrected some typographical and grammatical errors throughout the text and included in the legend of tables 2, 3, 4 and 5 the meaning of T0, T2, and T4.